# Analizing Teens an Analysis from the Perspective of Gamers in Youtube

**Raquel Lozano-Blasco [1,\*], M* Pilar Latorre-Martínez [2] and Alejandra Cortes-Pascual [3]**

[1] Department of Psychology and Sociology, Faculty of Humanities and Science Education, University of Zaragoza, 50001 Zaragoza, Spain

[2] Department of Business & Administration, University of Zaragoza, 50001 Zaragoza, Spain; latorrep@unizar.es

[3] Department of Sciences Education, Faculty of Education, University of Zaragoza, 50001 Zaragoza, Spain; alcortes@unizar.es

\* Correspondence: rlozano@unizar.es

**Abstract:** (1) Gamers are a new social phenomenon on YouTube whose success is based on their humour and social identity. The aim of this research is to deepen the understanding of the behaviour of the 100 gamers with the largest numbers of fans worldwide by studying their channels on YouTube; (2) Methods: This is a longitudinal research study from 20 August 2019 to 20 August 2020. The methodology consists of three techniques: social media analysis, opinion mining or sentiment analysis, and qualitative semantic analysis; (3) Results: The results of regression and KPI analysis confirm that the most popular contents have high levels of humour, positive polarity, irony, and subjectivity. In addition, the jargon of the digital community is used, focusing on group identification; (4) Conclusions: We conclude that teenagers use YouTube to search content that is cheerful, fun, and with high doses of humor and irony, in which gamers narrate their own vision of reality. Understanding these characteristics makes it possible to adapt educational channels to the interests of the adolescent community. At the same time, it allows us to understand how group identity is constructed in the virtual community, being able to establish lines of intervention from the educational and family orientation.

**Keywords:** gamers; YouTube; social networking sites

## 1. Introduction

It is well known that the excessive use of social networks can lead to changes in people's behaviour and neural structures [1–3]. Regardless of the risks involved in excessive use, 21st-century society demands competence in critically analyzing the information presented on social networks [4]. The use of screens results in both improvements in certain skills and damage other cognitive ones [5–7]. Some social networks such as Facebook and Twitter can help improve users' communication skills, while YouTube, whose users act more passively, does not imply any clear acquisition of skills [8–11]. YouTube is more than a website; it is a social network. This is because users or virtual communities gather on YouTube to share common interests, actively participating through "likes", "dislikes", and writing comments. That is, the community is not passive, but acquires a prosumer vision, the community consumes videos while generating new content [4,7,8].

However, YouTube has great educational potential, and it is worth asking what we can learn from gamers and their posts to engage students on this social network. The importance given by the adolescent population to remaining on social networks may explain the exponential growth in the time invested in this activity, currently being an average of two hours per day [12]. In this sense, it is necessary to start from the concept of a mobile-centric society [13], and the representation of reality from post-modernism where reality

and theatricality are confused [14]. However, as Gonzalez and Esteban [13] point out, the school has not been able to meet the challenge posed by new technologies and is characterised by a high degree of impermeability. In the terms of Prensky [15], the new generations of digital natives need more interactivity. However, the 19th century education system, with its sequential logic and exposure system, is no longer valid [15]. Social networks are structured as new relationship ecologies where a participatory digital culture emerges, in the terms of Jekins in Gonzalez and Esteban [13]. Despite the many efforts and resources allocated to the understanding of the many and varied economic, social, legal, and ethical aspects of the recent developments on the Internet, and their consequences for the individual and society at large (i.e., Managing Alternatives for Privacy, Property and Internet Governace Observatory, or The Internet Governance Forum), measuring its progress and success is both challenging and tricky. To date, there are no longitudinal studies that specifically investigate and debate the existing motivation related teenagers in social networks and the changes needed to set up an improved governance structure for the education innovation ecosystem.

In a bedroom culture, based on the individual democratisation of the adolescent vis-à-vis the adult [16], where families lack control over social networks, and knowing the existence of the volatility of the adolescent stage, as well as vulnerable environments [15,17], how can we know what adolescents consume the most? How can we observe and understand them?

The contents presented by YouTubers can be categorized into blogs, video games, 'unboxing' (presenting a product), and others [18]. It is necessary to clarify that gender differences and stereotypes also exist. As far as YouTubers are concerned, few women are considered gamers, instead representing the majority in other types of modalities such as routine videos and make-up tutorials [19–21]. In the specific case of gamers, it is important to explain how they are considered YouTubers, that is, users who present audiovisual content (videoblogs) on YouTube on different topics; in this case, how to overcome different phases of online games or criticisms by other users with high levels of humour [19,20,22]. In other words, the videos that are posted represent a personal narrative on the common theme of video games, although such communication presents certain peculiarities. First of all, a parasocial relationship is established between a gamer and their followers, that is, a bond is generated without requiring physical contact, thereby providing a feeling of intimacy, so that both feed back into social support within the network itself [23–26]. In other words, users feel that they are referents of the digital culture [27]. A sense of belonging and self-disclosure are key processes in development during adolescence and may lead to this type of relationship [28]. In this sense, the generation of digital natives who make up the digital community are born into an environment that generates a prefabricated idea of identity, i.e., they develop according to desirable canons [29]. Similarly, their notion of intimacy pushes them to develop deep relationships through digital applications [29]. Moreover, although the theory of uses and rewards and the theory of self-determination highlight the relevance of controlling relationships, content, presentation, and impressions, they have the etiology of 'fear of missing out' (FoMO) and nomophobia [30]. However, it should not be overlooked that adolescents present a social identity, as indicated by Tajfel's theory [31]. In this way, a series of processes operate to achieve group identification. This generates a digital community of followers of an influencer whose identification reinforces their self-esteem through identification [31–34]. However, this social process of identity construction usually occurs without family or school counselling, so that adolescents are fickle [35]. It is necessary for school guidance to provide families and teachers with the necessary tools to learn how to teach a healthy use of social networks [36,37]. In addition, it is important to be aware of the difficulties in conducting counselling sessions at this stage of development [17]. Likewise, it is necessary to learn what adolescents' conception of social networks is, how they present themselves, and what they value most [38]—in other words, to study their ecosystem from an open vision

that allows us to understand their reality, as adolescents seek to connect with others, in the terms of Bauman [39].

The communication the gamers establish is not natural. Several investigations highlight the peculiarity of their form of communication based on informal conversations among friends, in which the gamer treats their followers as close and trusted people. In the specific case of the gamer PewDiePie, it is observed how he constantly uses terms to communicate and refer to his followers as "bro", "just", "guy" and "think" [40]. Authors such as Aran-Ramspott et al. [27] state that the most relevant element of a YouTuber is their 'vis comica', that is, their capacity to innovate and surprise rather than the image of a brand they may represent. In other words, the gamers combine technical language with the language of digital culture while using terms of proximity, establishing a communication strategy based on feedback through comments [41–43].

Third, the obvious theatricality of the gamers, their use of humor and the jargon of digital culture lead to uniformity in its contents [23,44,45]. In terms of coherence, it is common to find an idealized vision based on a high degree of positivity and subjectivity in the polarity of feelings [46–48], although Ferchaud [19] suggests, in the case of video games, that the positivity rate is quite balanced between positivity, negativity, and neutrality. Contents with themes such as body image, self-expression, travel, digital culture, and 'startups' (emerging companies) are associated with positivity, those related to depression, loneliness, and real-world relationships have negative polarity, and those with their own identity and anxiety are neutral [49]. Nevertheless, it is important to qualify that these contents can present contradictory discourses due to the significant amounts of humour involved. As for the emotions represented, they are diverse, ranging from surprise to fear or happiness [40,50].

It is, therefore, crucial to understand the ecology of social groups or virtual communities on YouTube, what captures the attention of teenagers, how gamers manage to communicate with their community to generate social ties, and what we can learn and transfer to education from this type of communication. The aim of the study is to analyse the 100 most influential and followed gamers at an international level, studying the communication of their most popular publications, in order to determine which elements that manage to capture the attention of the adolescent virtual community.

## 2. Materials and Methods

### 2.1. Research Model and Procedure and Data Analysis

The research was developed in three phases (see Figure 1). The first part of the study corresponded to a social media analysis (SNA) methodology based on data mining. The monitoring of accounts was carried out using Fanpage Karma software from 20 August 2019 to 20 August 2020. The data analysis generated a considerable volume of data, materialized as Key Performance Indicators (KPIs): number of likes, number of retweets, commitment, and number of fans [51]. In this way, the 100 publications with the greatest numbers of 'likes' were identified.

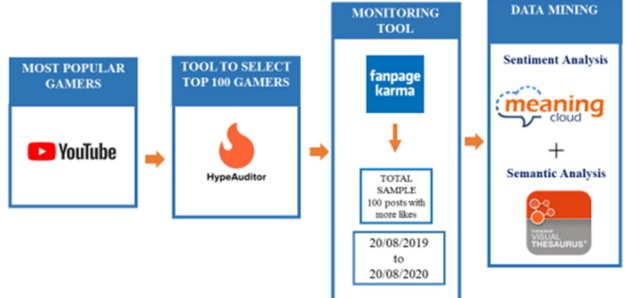

**Figure 1.** Research process from longitudinal study from 20 August 2019 to 20 August 2020 of the top 100 influencing gamers.

The second part was carried out using an opinion mining methodology based on the recognition of linguistic patterns through algorithms. In other words, a sentiment analysis [52] of the 100 publications with the largest numbers of 'likes' was conducted. Sentiment analyses study the emotional character of the messages emitted from natural language, providing a holistic vision of the new ecosystems generated on social networks [53–55]. The data analysis was executed using the MeaningCloud tool and the Emotion Recognition pack, enabling it to be executed in several languages in response to the linguistic diversity encountered.

The third part of the research corresponded to a qualitative methodology, in which a semantic analysis of the 100 most popular videos was carried out, materialized in the number of 'likes'. In other words, a semantic approach to natural language was applied in order to understand the themes dealt with in each video [56,57].

The data analysis consists of studying the frequency of words emitted, and this was studied by means of visual reports (in this case, word cloud graphics) as well as a network that allowed examination of the interactions of the most relevant terms with the ecosystem in question. This section was undertaken by means of the qualitative software Visual Thesaurus.

### 2.2. Research Context and Sample

The data acquisition was done using Fanpage Karma software from 20 August 2019 to 20 August 2020. The corpus was composed of 100 posts extracted from gamer accounts on the social network YouTube according to the index of 'I like' using the tool Fanpage Karma. Gamers were selected based on the index of subscribers available on the HypeAuditor platform. HypeAuditor is a marketing company specialized in creating influencer marketing campaigns. Its goal is to support agencies, brands, and platforms to improve effectiveness on social networks. In this way, it generates a series of rankings with different categories.

The 100 gamers with the most fans were selected. The inclusion criteria were (a) accounts aimed at teenagers or young people and (b) YouTubers deemed to be gamers. The sample is composed of 96% men and a scant 4% women. As for the distribution of nations the sample is composed of: Argentina (2%), Australia (4%), Brazil (15%), Canada (2%), Chile (1%), USA (24%), El Salvador (1%), India (2%), Indonesia (6%), Ireland (2%), Mexico (5%), Russia (7%), Saudi Arabia (2%), Spain (11%), Sweden (1%), Thailand (3%), Turkey (1%), Ukraine (1%), and the United Kingdom (9%). Regarding the majority typology of the videos, we find that 57% are "Let's Play Videos", 26% are "Live Stream Video", 9% are "Gaming Walkthrough", 3% are "Gaming Tutorials", 1% are "Game Analysis", 1% are "Game Reviews", 1% are "Preview an Upcoming Game", and the other 1% are "Secrets of the game". See Table A1.

### 2.3. Instrument Used and Their Validation

This section will explain the tools used to capture and interpret the development of the 100 YouTube channels.

First of all, the data of the different KPIs and main publications were re-analyzed using the Fanpage Karma software. This software allows the capture of key performance indicators or KPIs for each of the 100 accounts. The variables it studies (KPIs) are: (a) Number of comments ('Sum of direct comments (first level) and sub-comments (second level) on posts' by Fanpage Karma Academy), (b) fans ('Number of subscribers to a cannel' by Fanpage Karma Academy), (c) number of publications ('Number of videos published in the selected time period', Fanpage Karma Academy), (d) Number of Likes ('Average number of 'likes' on videos published in the selected period, divided by the number of videos in the selected period' by Fanpage Karma Academy), (e) Sum of the impressions of individual messages ('Number of views of videos published in the selected period' by Fanpage Karma Academy) [51].

Secondly, it is necessary to explain the importance of Meaning Cloud and Emotion Recognition. This software is specialized in analyzing the emotionality of language by means of algorithms [52–55]. In other words, a sentiment analysis is performed. Specifically, it analyzes the discourse of videos posted on YouTube by means of a series of variables: (a) polarity (an interpretation of whether the terms used in a message are very negative, negative, neutral, positive, very positive, or lacking emotionality), (b) agreement (messages categorized as neutral have an intermediate position between pleasant and unpleasant feelings; this variable shows whether there is really an absence of emotionality, i.e., the words are mostly neutral. In other words, it measures the agreement within the polarity), (c) irony (this provides classification according to the sarcasm or unintentionality of a message), (d) subjectivity (this establishes whether an opinion is expressed (subjective) or whether it explains or describes a fact (objective)), (e) confidence level (this shows the confidence value associated with the detected polarity), (f) emotion (the emotions included in Robert Plutchik's [58] theory of the 'emotion wheel', comprising joy, confidence, fear, surprise, sadness, aversion, anger, and anticipation), (g) emotion level (recorded intensity of each emotion; the higher the intensity, the higher the score).

Thirdly, it is necessary to specify the function of the Visual Thesaurus. This application is specialized in qualitative research and performs semantic analysis of content by generating word frequency, word clouds, and network graphics. In this way, it allows us to know which are the most influential concepts. This application provides a complementary vision that allows us to exemplify in written form what KPIs and sentiment analysis expose in a mathematical way.

The following research has followed ethical procedures both in the collection and processing of data and its management in accordance with the current regulations at the University of Zaragoza of the RGPD 2016/679 and the LO 3/2018 on the processing of personal data.

## 3. Results

### 3.1. KPI Analysis

The results of the KPI variables showed an average daily growth of 24.45% of the monitored accounts. In other words, the rate of capture of visualizations was very high. It is also interesting to note that the average value of the number of comments in these accounts was 1,765,839.84; the average number of publications per year was 350.01; the number of 'likes' was 28,811,393.3; and the number of channel views was 473,504,393. Similarly, the gamers studied have an average of 12,848,585.9 fans. The specific data of each channel studied are shown in Figure 2. As a summary, it should be noted that the gamer PewDiePie stood out above all others, followed by Vegetta777 and Fernanfloo. In this sense, it is worth indicating that most of the gamers studied were male, showing a significant gap between genders. On the other hand, the average of the 100 gamers in terms of KPI are: number of comments (mean value = 1,765,839.84); fans (mean value = 12,848,585.85); number of posts (median value = 350.01); number of likes (median value = 28,811,393.28); sum of impressions on individual posts (median value = 473,504,392.72). See Table A2 to see descriptive results of the KPIs of the study sample of the 100 most influential gamers ordered by followers. Regarding the longitudinal development of the KPIs of the most 'liked' publications, an interesting situation can be observed (see Figure 3). In the number of views, a significant peak is found in mid-February 2020. Let us recall that from February 16 to 24 the WHO issued a report establishing the imminent health crisis. On February 22, the number of views plummeted. This situation continued throughout the following months of the pandemic. Similarly, the number of comments plummeted in February. The community did not become active again until June 2020, when the Covid-19 emergency began to improve. Finally, regarding the number of likes,

we found that the pre-pandemic months were more active than the months of confinement and social distancing measures. There is a significant difference between the months prior to Covid-19 and the months of the pandemic.

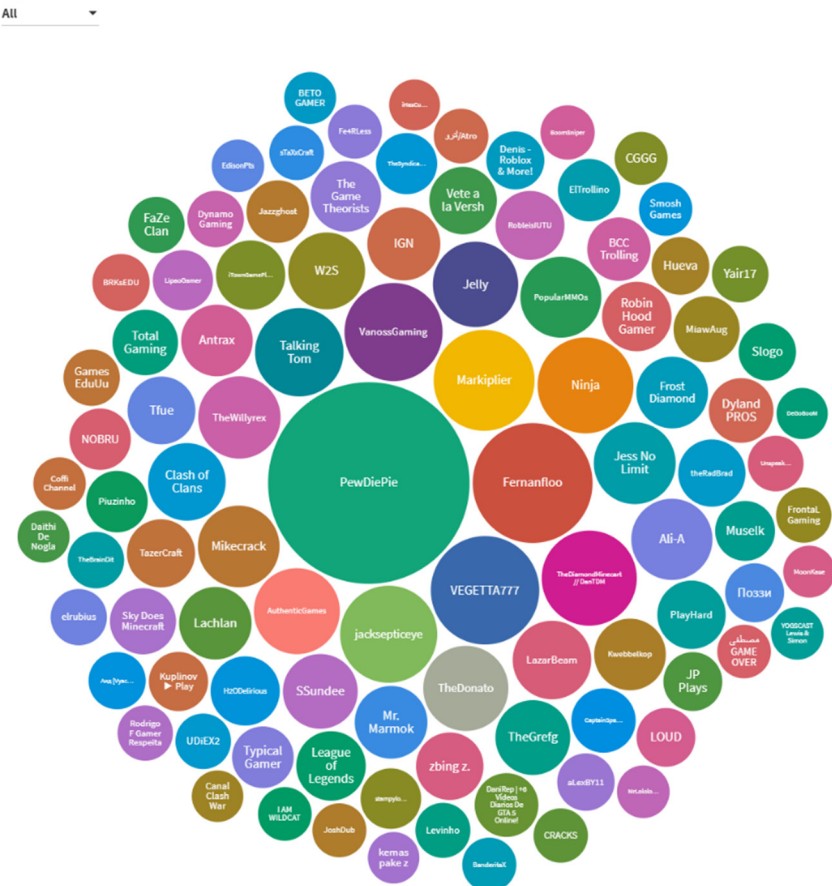

**Figure 2.** Packed interactive circles of top 100 gamer KPIs from 20 August 2019 to 20 August 2020. Exposed: https://public.flourish.studio/visualisation/4599936/ (accessed on 24 October 2020).

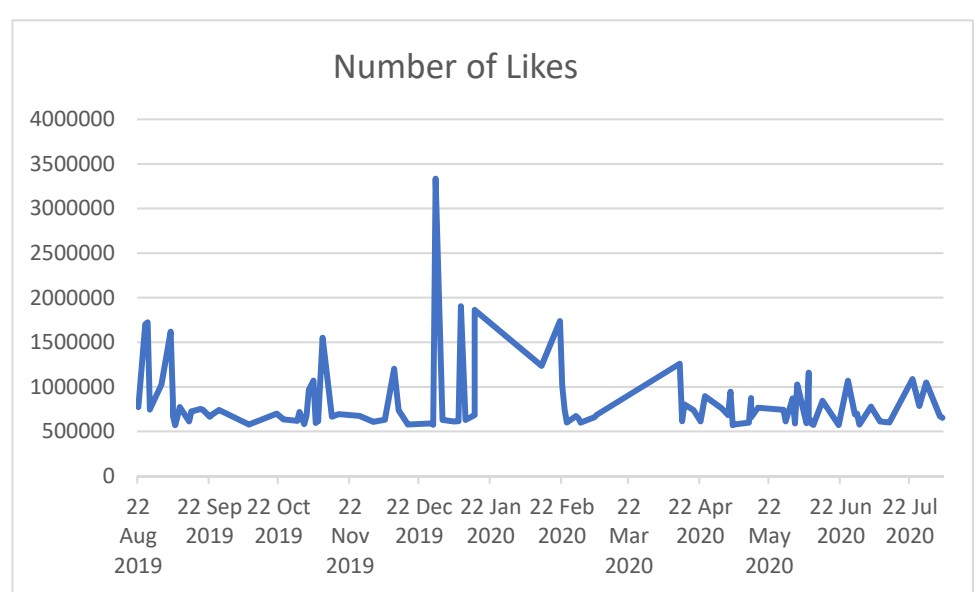

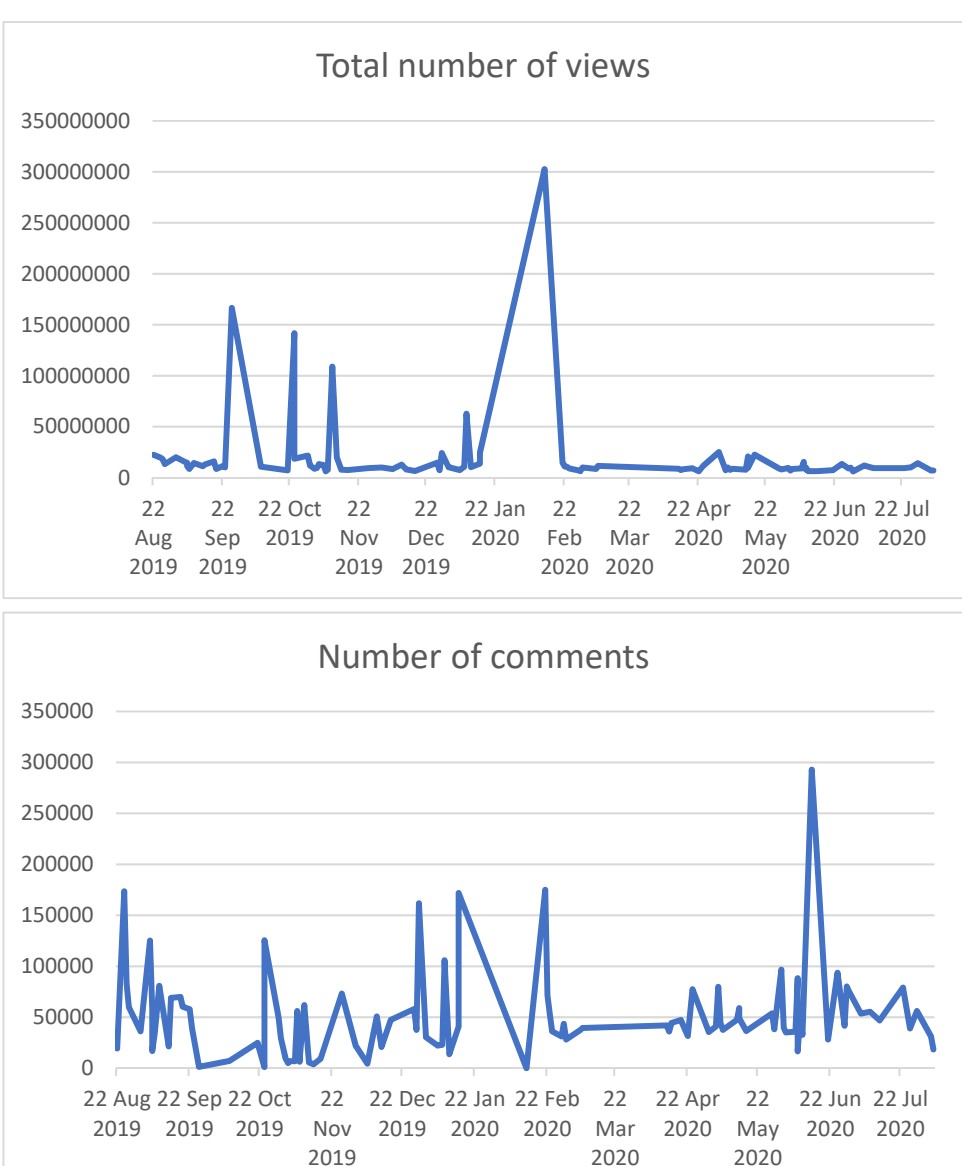

**Figure 3.** Longitudinal development of KPIs from 20 August 2019 to 20 August 2020.

*3.2. Sentiment Analysis*

The sentiment analysis returned interesting results with a high degree of confidence (77.97%) with regard to polarity. Initially, all messages were categorized as subjective, i.e., expressing an opinion and not describing or announcing a fact. Similarly, the rate of emotionality was 98%, in most cases stating messages with a significant emotional charge. In agreement, 69% of the publications were categorized as ironic, being able to establish how this element is transcendental. Consequently, the polarity of the sample was mostly positive, a not insignificant percentage of the publications were considered neutral and messages with a negative or unpleasant charge represented the minority. As for the emotions expressed, as shown in Figure 4, the majority were positive or pleasant, the most frequent being happiness, followed by trust. Anger and sadness were the only unpleasant emotions represented. In other words, there was an over-representation of the emotion happiness in the speech, which could be associated with high levels of humour or a sweetened vision of reality.

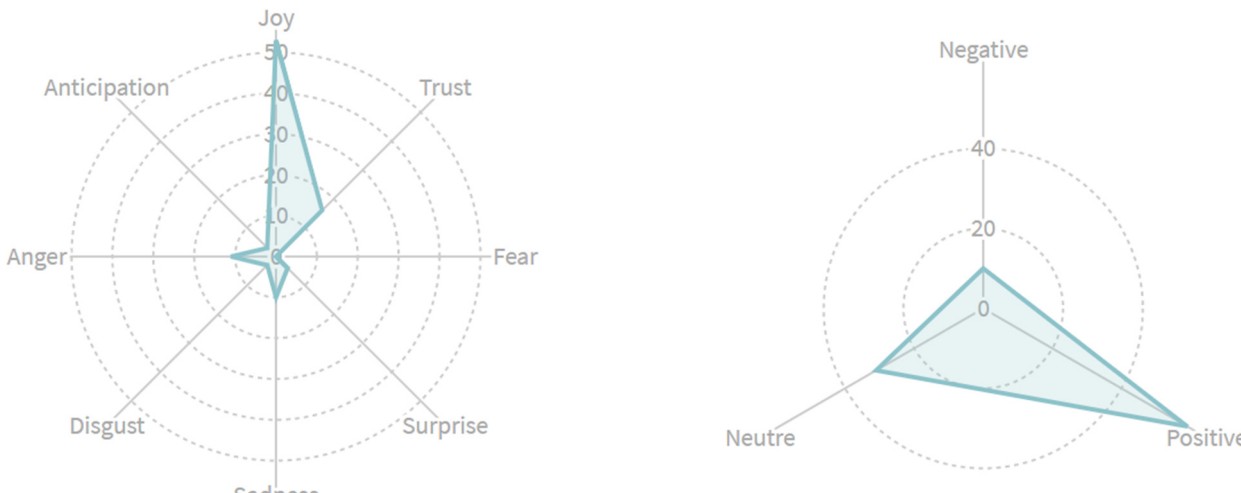

**Figure 4.** Polarity and emotions in the 100 publications with the greatest impact from 20 August 2019 to 20 August 2020 of the top 100 influencing gamers.

However, the polarity of the publications deserved further analysis. Thus, an analysis of Pearson's correlations was carried out (Table 1), revealing significant relationships among the study variables. With the purpose of understanding in greater measure these relations, those variables with significant results were selected.

**Table 1.** Pearson's correlations on the sentiment analysis (emotion, emotion level, polarity, agreement; confidence level, irony) in the 100 publications with the greatest impact.

| Variable | | Emotion | Emotion Level | Polarity | Agreement | Confidence Level | Irony |
|---|---|---|---|---|---|---|---|
| 1. Emotion [a] | Pearson's r | — | | | | | |
| | *p*-value | — | | | | | |
| 2.Emotion Level [b] | Pearson's r | 0.453 | — | | | | |
| | *p*-value | <0.001 | — | | | | |
| 3. Polarity [c] | Pearson's r | 0.750 | 0.405 | — | | | |
| | *p*-value | <0.001 | <0.001 | — | | | |
| 4. Agreement [d] | Pearson's r | 0.231 | 0.083 | 0.317 | — | | |
| | *p*-value | 0.021 | 0.411 | 0.001 | — | | |
| 5. Confidence Level [e] | Pearson's r | −0.077 | −0.305 | −0.146 | −0.373 | — | |
| | *p*-value | 0.447 | 0.002 | 0.148 | <0.001 | — | |
| 6. Irony [f] | Pearson's r | 0.075 | 0.288 | 0.120 | 0.218 | −0.950 | — |
| | *p*-value | 0.456 | 0.004 | 0.233 | 0.029 | <0.001 | — |

[a] Emotion: Robert Plutchik's theory of the Emotion Wheel. [b] Emotion Level: intensity of each emotion. [c] Polarity: an interpretation of whether the terms used in a message are very negative, negative, neutral, positive, very positive, or lacking emotionality. [d] Agreement: messages categorized as neutral have an intermediate position between pleasant and unpleasant feelings. [e] Confidence level: associated with the detected polarity. [f] Irony: this provides classification according to the sarcasm or unintentionality of a message.

Then, four multiple regression models were made using a forward introduction method (Table 2) that allowed the polarity (dependent variable) to be examined. The choice of forward step method to use was based on the need to determine which factor was the most relevant in polarity of publications. This decision was taken to prevent contamination among the variables and favouring the exclusion of those that were irrelevant.

**Table 2.** Regression models through steps forward on polarity (dependent variable).

| Modelo | | Unstandardized | Standard Error | Standardized | t | *p* | R² | RMSE | F | *p* |
|---|---|---|---|---|---|---|---|---|---|---|
| 1 | (Intercept) | −0.32 | 2.99 | | −0.10 | 0.91 | 0.59 | 0.44 | 27.14 | <0.001 |
| | Emotion | 0.19 | 0.02 | 0.67 | 8.69 | <0.001 | | | | |
| | Emotion level | 0.00 | 0.00 | 0.08 | 1.04 | 0.29 | | | | |
| | Agreement | 0.72 | 0.40 | 0.15 | 1.78 | 0.07 | | | | |
| | Confidence level | −0.00 | 0.03 | −0.00 | −0.01 | 0.99 | | | | |
| | Irony | 0.01 | 0.35 | 0.01 | 0.04 | 0.96 | | | | |
| 2 | (Intercept) | −0.35 | 0.31 | | −1.12 | 0.26 | 0.59 | 0.44 | 34.29 | <0.001 |
| | Emotion | 0.19 | 0.02 | 0.67 | 8.91 | <0.001 | | | | |
| | Emotion level | 0.00 | 0.00 | 0.08 | 1.07 | 0.28 | | | | |
| | Agreement | 0.72 | 0.33 | 0.15 | 2.18 | 0.03 | | | | |
| | Irony | 0.01 | 0.10 | 0.01 | 0.17 | 0.86 | | | | |
| 3 | (Intercept) | −0.35 | 0.31 | | −1.13 | 0.25 | 0.59 | 0.43 | 46.18 | <0.001 |
| | Emotion | 0.19 | 0.02 | 0.67 | 9.00 | <0.001 | | | | |
| | Emotion level | 0.00 | 0.00 | 0.08 | 1.18 | 0.23 | | | | |
| | Agreement | 0.73 | 0.32 | 0.15 | 2.29 | 0.02 | | | | |
| 4 | (Intercept) | −0.30 | 0.31 | | −0.97 | 0.33 | 0.58 | 0.43 | 68.27 | <0.001 |
| | Emotion | 0.20 | 0.01 | 0.71 | 10.63 | <0.001 | | | | |
| | Agreement | 0.72 | 0.32 | 0.15 | 2.25 | 0.02 | | | | |

Multiple regression generated four explanatory models of videos' polarity. The strongest from the statistical point of view is model 4. Model 4 explains how 57% of the polarity (R² = 0.57, *p* < 0.001) is explained by emotion and agreement significantly (*p* < 0.001). That is, the polarity of the videos published by gamers is explained by the amount of emotional words they use. In addition, the parameters F (F = 68.27) and t (emotion t = 10.63; agreement t = 2.25) show a positive relationship. That is, the dependent variable of polarity grows together with the emotion and agreement variables.

In other words, gamers' discourse has a high prevalence of terms that convey emotions. That is, it is not a speech that describes a situation. It is a conversation where emotions, especially joy, are expressed and reflected. It is necessary to remember how this is the most prevalent emotion in the publications (see Figure 3).

*3.3. Semantic Analysis*

The semantic analysis was composed of 54,321 words, mainly from the English language. The occurrence of words indicated that the most used were 'like' (365), 'get' (335), 'have' (319), 'just' (285), 'but' (265), 'can' (217), 'know' (212), 'all' (203), 'okay' (195), and 'one' (189). In Figures 5a and A1, these words are presented related to others, generating ramifications. As the interrelationship continues, more density graphs were generated (see Figure 5b,c).

**(a)**

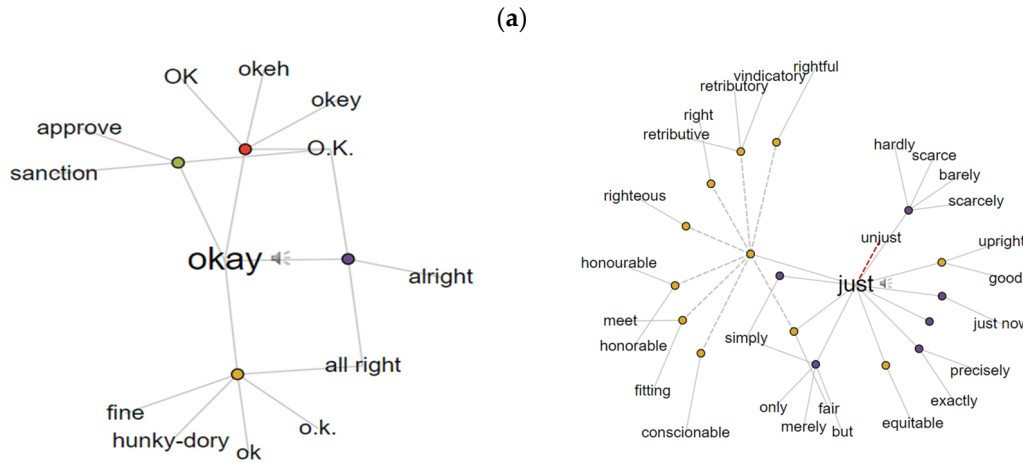

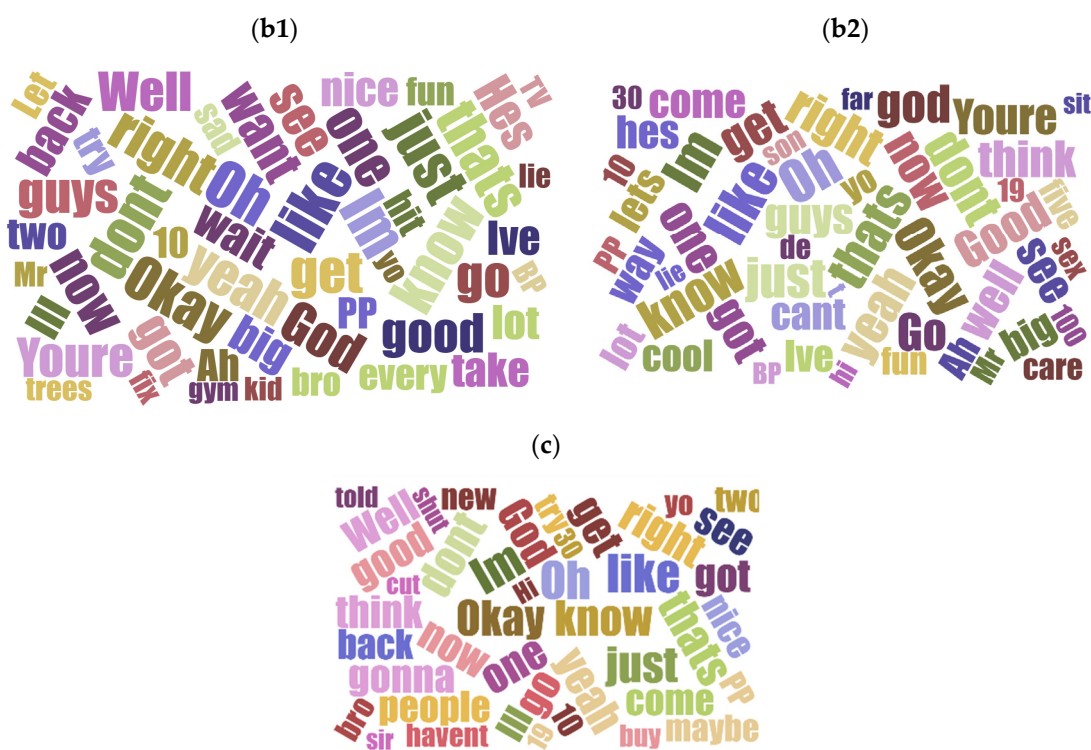

**Figure 5.** Semantic analysis of the 100 videos with the largest numbers of likes from 20 August 2019 to 20 August 2020 of the top 100 influencing gamers. (**a**) Branch graphics. (**b1**) Density clouds August 2019–February 2020. (**b2**) March 2020 to August 2020. (**c**) Density clouds with 500 most frequent words.

Their interrelationship shows active communication, directed towards other people, seeking to establish a conversation using expressions typical of digital jargon, such as 'guys' or 'bro' (see Figure 5b,c). In addition, we observe a series of verbs that favor the phatic or expressive function of language, whose purpose is to ensure or maintain communication between the sender (gamers) and the receiver (virtual community). That is why we find digital slang expressions such as 'guys' or 'bro' that favor the feeling of community and group. Two of the most frequent words are 'just' and 'okay'. Both are expressions that seek to maintain and ensure communication between sender (gamer) and receiver (followers). Thus, we find that "okay" is related to other terms with the same function such as "approve", "OK", "okeh", "all right", and "fine", and "just" is related to concepts such as "exactly", "precisely", "good", "up right", and "simply". In turn, we find an important branch that relates "just" and "simply" to terms such as "rightful", "vindicator", "retributory", "honourable", "honorable".

Likewise, adjectives and adverbs whose meanings are related to positive aspects (such as 'honourable', 'right', 'good', 'exactly', 'totally', 'completely', and 'extraordinary') were common, showing agreement or reaffirming an idea (see Figure A1). The distribution of these words according to the word density graphs reveals how the most used terms were action verbs that seek interaction with followers, for instance 'look', 'cool', and 'yeah'; that is to say, these words correspond to a monologue-like narration aimed at a very specific audience. Thus, the visualizations with the highest 'I like' rates were those in which the gamer sought the interaction of their followers using terms that have a positive connotation and show agreement.

This type of conversation is reminiscent of post-modern theater, where fiction is sought to reach reality. In this way, what happens on the screen (in the YouTube video) is not only a content to watch passively, as, for example, television is. The conversation developed by the influencer, through group unit terms 'cool' and 'bro', together with terms such as 'okay' and 'just' that guarantee the open channel of communication, make this

content feel real. In other words, the gamer manages to establish a parasocial relationship with his followers, developing a delayed conversation and getting the virtual community to participate actively through the option of 'like', 'dislike', and 'comments'. Finally, no differences are observed in the word clouds of videos made from August 2019 to February 2020 (pre-pandemic period of Covid-19) to videos published between March and August 2020. The communication maintains the same terminology.

## 4. Discussion

The widespread use of the YouTube social network is evident from the results returned from the KPIs. In this sense, this study's findings are congruent with previous research pointing to the extended use of this platform among the adolescent and young population [59,60]. On the other hand, the investigation agrees with Ferchaud et al. [19–21] on the scarcity of female representation among the most influential gamers at the international level.

The results of the polarity show some inherent and typical features of the new digital culture. The high rates of positivity together with the significant levels of subjectivity identified are related to the sweetened vision of the world highlighted in previous research [46–48], especially in content focusing on everyday life, travel, and games [49]. This study's results are consistent with such research. Nevertheless, it is necessary to indicate that the data presented in this study are not completely in agreement with the work of Ferchaud et al. [19], who in the specific case of videos about video games found a negative polarity rate of 20%, above that recorded in the present research. On the other hand, it is necessary to reaffirm the results of previous studies such as that of Ashman et al. [23] carried out with the adolescent population. The contents that this population has been found to like the most are of a humorous type, the predominant emotion being happiness. The data presented in this research follow the same line as these previous authors. By contrast, studies by Beers Fägersten [40] and Dewaele [50] demonstrate greater emotional diversity, ranging from surprise to fear and happiness. These results are consistent with the post-modernist concept of social identity or identity establishment [14], where reality and fantasy intermingle.

In sum, the results of the semantic analysis have exposed the existence of words loaded with positive emotions that seek interaction with followers. We agree with Scheinbaum [45] that these gamers do not present a transgressive image, but follow the fashion of the moment, so that the most used words present a certain uniformity. In the first place, it has been found that they are mostly verbs of action and social interaction, necessary for gamers' humorous narration, in line with previous investigations pointing out how YouTubers oscillate between manifesting expert knowledge of digital culture and using colloquial aspects that remind their audiences of conversations between friends [27,40]. The results of the present study are similar to those found by Beers Fägersten [40] on PewDieDie, highlighting terms such as 'bro', 'just', 'guy', and 'think', which this gamer uses to communicate with his followers. In the same way, the use of such close and direct language appears to be an essential strategy in recruiting followers, with appealing for interaction and referencing one's followers in videos coinciding with the research carried out by Pereira et al. [18]. In this way, we find how the adolescent virtual community positively reaffirms those traits that confer a social identity [31–34].

This type of communication seeks to generate an affective bond and emotional commitment, a parasocial relationship between gamer and follower, in line with previous research [20,23–26,40]. An example of this close and emotional communication is the digital natives' conception of intimacy, which pushes them to develop close relationships through new technologies [29]. Similarly, there is agreement on how this communication strategy is mediated by feedback through 'likes' and commentary [41–43]. In this sense, we agree with Bérail et al. [24] and Hartmann [20] that either current rates of loneliness and social difficulties are stimulating adolescents and young adults to interact face-to-face less and less than in the past, or we are facing a new phenomenon of friendships and

relationships, in which users feel encouraged to be part of the digital culture of the Internet [27], as explained by their contemporary sense and self-disclosure [28].

This study is not without its limitations. First, the context of the pandemic in which the research was conducted must be taken into account, i.e., it is a temporary photo. Our results show how Covid-19 modified the behavior of the digital community. Since the WHO statement in late February and early March 2020, the digital community stopped paying attention to these channels. This can be seen in Figure 3, which shows a drop in the number of 'likes', number of views and comments. However, gamers continued to make the same type of videos. It is, therefore, necessary to repeat the study in the following years 2021 and 2022 to determine the behavior of the digital community. Second, this study has focused on the particular case of gamers, but research on other profiles of influencers is needed. At this point, it is necessary to clarify how there is an important gender gap and so it is essential to carry out research into the establishment of gender stereotypes among influencers. On the other hand, it has not been possible to find studies with the same methodology that would afford detailed comparison due to the speed with which new social networks are established. It would be interesting to repeat this study after a few years to analyze if there have been any changes in behaviour. Similarly, it would be worth carrying out a study with a survey design to analyze the perceptions of the adolescent and young adult population about the new manifestation of gamers.

## 5. Conclusions

In response to the research question, the teenage digital community prefers those communications in which the influencer uses a popular jargon, typical of the digital community, a relaxed discourse, in which humour and high levels of positivity are dealt with, as well as a subjective and ironic vision of the gamer himself. These traits are what determine the success of a publication among the adolescent digital community. In this sense, a practical application in the educational world is the generation or adaptation of learning channels to these characteristics. In this sense, videos should be short, fun, cheerful, with a high dose of humour and with a relaxed language in line with the digital world where it is developed, recognising its essence and relevance. Perhaps in this way, YouTube would become a learning-friendly platform like Twitter or Facebook [8–11].

Similarly, knowledge of their jargon, and of the type of publications that are most consumed by this generation, makes educational guidance intervention possible. This is due, firstly, to the actual knowledge of the preferred publication typology. Secondly, adaptation to their jargon implies a recognition of their group identity and shows interest and respect for it, essential elements for efficient adolescent–family or adolescent–teacher communication [17,36,37]. Likewise, the strong difference between men and women who have more consuming profiles should be the subject of debate and critical treatment at school.

In conclusion, gamers are an established phenomenon in the digital community and the most influential positions are occupied by men, whereas only a relatively minor role is played by women. Moreover, the videos that enjoy the most popularity are those in which the gamer expresses an opinion, loaded with subjectivity and emotion, thereby transmitting a happy and joyful vision accompanied by high levels of humour. Accordingly, it should be specified how gamers' archetypal discourse is constructed by seeking direct interaction with their followers through the use of jargon and a context of closeness similar to a conversation between friends.

It is clear from the above that the motivation of adolescents corresponds to feeling part of the digital community and that digital media underlie as a culture and not merely as a communication tool.

As we mentioned earlier, there are no studies that specifically investigate the motivation related teenagers in social networks and the changes needed to set up an improved educational governance. We strongly believe that initiatives inspire and support the education innovation ecosystem and are also linked to different approaches and visions by

scholars, public administrations, practitioners, and why not, companies and social networks are needed. For example, the creation of the International Observatory of Adolescent Use of Social Networks could be key to addressing the future challenges and limiting the risks of developments on the Internet as a cultural space and a space for creating social value.

**Author Contributions:** Conceptualization, R.L.-B., M.P.L.-M. and A.C.-P.; methodology, R.L.-B.; formal analysis, R.L.-B.; investigation, R.L.-B.; writing—original draft preparation and review and editing, R.L.-B. All authors have read and agreed to the published version of the manuscript.

**Funding:** This research was funded by EDUCAVIVA research group by Zaragoza University.

**Institutional Review Board Statement:** Not applicable for studies not involving humans or animals. The following research has followed ethical procedures both in the collection and processing of data and its management in accordance with the current regulations at the University of Zaragoza of the RGPD 2016/679 and the LO 3/2018 on the processing of personal data. The study was conducted according to the guidelines of the Declaration of Helsinki.

**Informed Consent Statement:** Not applicable for studies not involving humans or animals. The following research has followed ethical procedures both in the collection and processing of data and its management in accordance with the current regulations at the University of Zaragoza of the RGPD 2016/679 and the LO 3/2018 on the processing of personal data. The study was conducted according to the guidelines of the Declaration of Helsinki.

**Data Availability Statement:** Not applicable.

**Acknowledgments:** This research is supported by a contract coverage of the Ministry of Science, Innovation and Universities of Spain (*Formacion de Profesorado Universitario*—FPU).

**Conflicts of Interest:** The authors declare no conflict of interest.

## Appendix A

**Table A1.** Description of the study sample of the 100 most influential gamers: gender, country, language, and channel description.

| YouTube Profile | Gender of Influencer | Country | Language | Description Channel |
|---|---|---|---|---|
| PewDiePie | Male | EEUU | English | Let's Play Videos |
| Fernanfloo | Male | El Salvador | Spanish | Live Stream Video |
| VEGETTA777 | Male | Spain | Spanish | Game Reviews |
| Markiplier | Male | EEUU | English | Let's Play Videos |
| VanossGaming | Male | Canada | English | Let's Play Videos |
| jacksepticeye | Male | Ireland | English | Gaming Walkthrough |
| Ninja | Male | EEUU | English | Live Stream Video |
| TheDiamondMinecart//DanTDM | Male | United Kingdom | English | Let's Play Videos |
| Talking Tom | Male | EEUU | English | Let's Play Videos |
| AuthenticGames | Male | Brasil | Portuguese | Gaming Tutorials |
| Jelly | Male | United Kingdom | English | Live Stream Video |
| TheDonato | Male | Argentina | Spanish | Let's Play Videos |
| TheWillyrex | Male | Spain | Spanish | Let's Play Videos |
| Jess No Limit | Male | Indonesia | English | Gaming Tutorials |
| Mikecrack | Male | Spain | Spanish | Let's Play Videos |
| Ali-A | Male | United Kingdom | English | Gaming Walkthrough |
| PopularMMOs | Male | EEUU | English | Let's Play Videos |
| LazarBeam | Male | Australia | English | Let's Play Videos |
| Clash of Clans | Male | EEUU | English | Let's Play Videos |
| W2S | Male | United Kingdom | English | Live Stream Video |
| SSundee | Male | EEUU | English | Let's Play Videos |
| TheGrefg | Male | Spain | Spanish | Live Stream Videos |
| IGN | Male | EEUU | English | Preview an Upcoming Game |

| | | | | |
|---|---|---|---|---|
| Mr. Marmok | Male | Russia | Russian | Gaming Walkthrough |
| Lachlan | Male | Australia | English | Live Stream Videos |
| Antrax | Male | Mexico | Spanish | Live Stream Videos |
| Frost Diamond | Male | Indonesia | English | Let's Play Videos |
| Kwebbelkop | Male | EEUU | English | Let's Play Videos |
| The Game Theorists | Male | EEUU | English | Secrets of the game |
| League of Legends | Male | EEUU | English | Let's Play Videos |
| Robin Hood Gamer | Male | Brasil | Portuguese | Let's Play Videos |
| H₂ODelirious | Male | EEUU | English | Gaming Walkthrough |
| iTownGamePlay ★Terror&Diversión★ | Male | Spain | Spanish | Let's Play Videos |
| RobleisIUTU | Male | Argentina | Spanish | Live Stream Videos |
| PlayHard | Male | Brasil | Portuguese | Gaming Tutorials |
| TazerCraft | Male | Brasil | Portuguese | Let's Play Videos |
| Tfue | Male | EEUU | English | Live Sream Videos |
| Vete a la Versh | Male | Mexico | Spanish | Let's Play Videos |
| zbing z. | Female | Thailand | English | Let's Play Videos |
| theRadBrad | Male | EEUU | English | Gaming Walkthrough |
| MiawAug | Male | Indonesia | English | Gaming Walkthrough |
| Sky Does Minecraft | Male | EEUU | English | Let's Play Videos |
| Total Gaming | Male | India | English | Live Stream Videos |
| Dyland PROS | Male | Indonesia | English | Let's Play Videos |
| CaptainSparklez | Male | EEUU | English | Let's Play Videos |
| DaniRep|+6 Vídeos Diarios De GTA 5 Online! | Male | Spain | Spanish | Live Stream Videos |
| BCC Trolling | Male | United Kingdom | English | Live Stream Videos |
| ElTrollino | Male | Spain | Spanish | Let's Play Videos |
| Jazzghost | Male | Brasil | Portuguese | Let's Play Videos |
| Typical Gamer | Male | Canada | English | Gaming Walkthrough |
| Piuzinho | Male | Brasil | Portuguese | Live Stream Videos |
| NOBRU | Male | Brasil | Portuguese | Let's Play Videos |
| TheSyndicateProject | Male | United Kingdom | English | Gaming Walkthrough |
| stampylonghead | Male | United Kingdom | English | Gaming Walkthrough |
| LipaoGamer | Male | Brasil | Portuguese | Let's Play Videos |
| Muselk | Male | Australia | English | Live Stream Videos |
| Kuplinov ► Play | Male | Russia | Russian | Live Stream Video Games |
| Поззи | Male | Russia | Russian | Let's Play videos Games |
| JP Plays | Male | Brasil | Portuguese | Let's Play videos Games |
| LOUD | Male | EE.UU | English | Let's Play videos Games |
| Denis-Roblox & More! | Male | EE.UU | English | Let's Play videos Games |
| Hueva | Male | Mexico | Spanish | Live Stream Video Games |
| aLexBY11 | Male | Spain | Spanish | Live Stream Video Games |
| Slogo | Male | United Kingdom | English | Let's Play videos Games |
| BRKsEDU | Male | Brasil | Portuguese | Let's Play videos Games |
| Аид [VyacheslavOO] | Male | Russia | Russian | Let's Play videos Games |
| Yair17 | Male | Mexico | Spanish | Let's Play videos Games |
| Dynamo Gaming | Male | India | English | Let's Play videos Games |
| TheBrainDit | Male | Ukraine | Ukrainian | Let's Play videos Games |
| Games EduUu | Male | Brasil | Portuguese | Let's Play videos Games |
| elrubius | Male | Spain | Spanish | Live Stream Video Games |
| FaZe Clan | Male | EE.UU | English | Let's Play videos Games |
| Unspeakable | Male | EE.UU | English | Live Stream Video Games |
| UDiEX2 | Male | Thailand | Thai | Live Stream Video Games |
| FrontaL Gaming | Male | indonesia | English | Live Stream Video Games |
| Rodrigo F Gamer Respeita | Male | Brasil | Portuguese | Let's Play videos Games |
| Levinho | Male | Sweden | Swedish | Let's Play videos Games |
| أترو/Atro | Male | Saudi Arabia | Arab | Live Stream Video Games |
| sTaXxCraft | Male | Spain | Spanish | Let's Play videos Games |
| CRACKS | Male | Spain | Spanish | Let's Play videos Games |

| | | | | |
|---|---|---|---|---|
| BoomSniper | Male | Mexico | Spanish | Let's Play videos Games |
| BanderitaX | Male | Saudi Arabia | Arab | Let's Play videos Games |
| JoshDub | Male | Australia | English | Let's Play videos Games |
| Fe4RLess | Female | EE.UU | English | Let's Play videos Games |
| I AM WILDCAT | Male | EE.UU | English | Live Stream Video Games |
| مصطفىGAME OVER | Male | Turkey | turkish | Let's Play videos Games |
| Smosh Games | Male | EE.UU | English | Game Analysis |
| CGGG | Male | Thailand | Thai | Let's Play videos Games |
| MrLololoshka (Роман Фильченков) | Male | Russia | Russian | Let's Play videos Games |
| YOGSCAST Lewis & Simon | Male | United Kingdom | English | Let's Play videos Games |
| Coffi Channel | Male | Russia | Russian | Let's Play videos Games |
| EdisonPts | Male | Russia | Russian | Live Stream Video Games |
| Daithi De Nogla | Male | Irland | English | Live Stream Video Games |
| MoonKase | Female | Brasil | Portuguese | Let's Play videos Games |
| BETO GAMER | Male | Brasil | Portuguese | Let's Play videos Games |
| Canal Clash War | Male | Brasil | Portuguese | Let's Play videos Games |
| kemas pake z | Male | Indonesia | English | Live Stream Video Games |
| DeGoBooM | Male | Chile | Spanish | Let's Play videos Games |
| iHasCupquake | Female | EE.UU | English | Let's Play videos Games |

**Table A2.** Descriptive results of the KPIs of the study sample of the 100 most influential gamers ordered by followers.

| YouTube Profile | Number of Comments (Total) [a] | Fans [b] | Number of Publications [c] | Number of Likes [d] | Sum of the Impressions of Individual Messages [e] |
|---|---|---|---|---|---|
| PewDiePie | 12,034,046 | 106,000,000 | 313 | 163,926,899 | 2,326,202,351 |
| Fernanfloo | 752,957 | 37,500,000 | 4 | 5,910,193 | 40,932,718 |
| VEGETTA777 | 1,647,323 | 30,700,000 | 667 | 97,453,430 | 997,268,571 |
| Markiplier | 2,917,568 | 26,600,000 | 285 | 35,456,085 | 792,006,032 |
| VanossGaming | 196,334 | 25,100,000 | 182 | 14,017,478 | 432,004,572 |
| jacksepticeye | 4,193,100 | 24,600,000 | 395 | 41,381,254 | 788,272,256 |
| Ninja | 660,423 | 24,000,000 | 254 | 11,961,429 | 306,558,138 |
| TheDiamondMinecart//Dan TDM | 2,982,603 | 23,600,000 | 197 | 15,043,432 | 563,737,951 |
| Talking Tom | 20,412 | 20,400,000 | 37 | 5,182,713 | 1,323,594,769 |
| AuthenticGames | 1,059,443 | 19,300,000 | 333 | 18,652,417 | 281,341,564 |
| Jelly | 2,541,435 | 19,000,000 | 699 | 115,652,114 | 1,918,231,945 |
| TheDonato | 4,754,133 | 18,900,000 | 328 | 114,147,981 | 1,114,369,195 |
| TheWillyrex | 676,915 | 17,700,000 | 376 | 31,571,403 | 422,420,103 |
| Jess No Limit | 21,829,800 | 17,500,000 | 570 | 46,088,980 | 883,955,761 |
| Mikecrack | 2,411,291 | 17,400,000 | 138 | 29,737,381 | 846,339,479 |
| Ali-A | 1,084,709 | 17,200,000 | 248 | 14,763,289 | 409,162,441 |
| PopularMMOs | 1,110,049 | 17,100,000 | 135 | 4,706,184 | 156,173,234 |
| LazarBeam | 2,921,237 | 16,100,000 | 127 | 39,976,290 | 1,309,433,268 |
| Clash of Clans | 149,864 | 15,700,000 | 64 | 3,675,023 | 326,604,955 |
| W2S | 75,589 | 15,400,000 | 9 | 1,893,082 | 50,531,104 |
| SSundee | 3,819,036 | 14,600,000 | 321 | 71,537,986 | 1,070,835,104 |
| TheGrefg | 3,418,441 | 14,600,000 | 271 | 72,950,917 | 923,059,872 |
| IGN | 3,160,339 | 13,900,000 | 4625 | 23,074,786 | 995,606,769 |
| Mr. Marmok | 1,243,972 | 13,800,000 | 33 | 19,593,877 | 245,807,086 |
| Lachlan | 794,288 | 13,700,000 | 157 | 13,772,185 | 558,834,945 |
| Antrax | 1,305,993 | 13,400,000 | 75 | 18,021,754 | 250,400,504 |
| Frost Diamond | 9,519,299 | 13,400,000 | 413 | 46,201,797 | 802,789,236 |
| Kwebbelkop | 438,449 | 13,400,000 | 352 | 19,812,635 | 687,749,179 |

| | | | | | |
|---|---|---|---|---|---|
| The Game Theorists | 1,256,316 | 13,000,000 | 73 | 10,015,294 | 231,114,925 |
| League of Legends | 685,955 | 12,600,000 | 109 | 12,330,025 | 516,903,207 |
| Robin Hood Gamer | 1,339,743 | 12,600,000 | 475 | 59,633,254 | 853,764,104 |
| H₂ODelirious | 652,499 | 12,500,000 | 337 | 13,703,823 | 279,414,625 |
| iTownGamePlay ★Terror&Diversión★ | 780,326 | 12,500,000 | 596 | 6,796,971 | 97,734,140 |
| RobleisIUTU | 2,284,084 | 12,400,000 | 292 | 38,234,922 | 512,024,360 |
| PlayHard | 875,272 | 12,200,000 | 286 | 36,169,691 | 305,098,805 |
| TazerCraft | 378,164 | 12,100,000 | 198 | 10,827,996 | 145,934,305 |
| Tfue | 304,257 | 12,000,000 | 97 | 6,872,164 | 170,587,819 |
| Vete a la Versh | 80,193 | 11,900,000 | 28 | 3,060,509 | 45,111,876 |
| zbing z. | 791,845 | 11,900,000 | 366 | 13,695,613 | 625,001,678 |
| theRadBrad | 793,946 | 11,800,000 | 521 | 9,974,510 | 293,265,167 |
| MiawAug | 4,248,846 | 11,400,000 | 423 | 26,085,893 | 769,930,306 |
| Sky Does Minecraft | 111,734 | 11,400,000 | 203 | 1,071,615 | 12,795,664 |
| Total Gaming | 2,573,235 | 11,200,000 | 713 | 91,165,996 | 1,156,491,047 |
| Dyland PROS | 5,463,883 | 10,900,000 | 406 | 17,260,663 | 316,178,164 |
| CaptainSparklez | 335,183 | 10,800,000 | 320 | 4,202,096 | 103,174,162 |
| DaniRep\|+6 Vídeos Diarios De GTA 5 Online! | 691,338 | 10,600,000 | 636 | 19,390,712 | 431,549,582 |
| BCC Trolling | 437,524 | 10,500,000 | 373 | 9,481,413 | 364,709,704 |
| ElTrollino | 564,603 | 10,200,000 | 65 | 13,667,484 | 551,701,559 |
| Jazzghost | 2,820,931 | 10,200,000 | 631 | 42,332,135 | 470,739,796 |
| Typical Gamer | 234,974 | 9,910,000 | 414 | 7,399,989 | 364,029,865 |
| Piuzinho | 1,323,554 | 9,870,000 | 80 | 30,047,725 | 153,661,518 |
| NOBRU | 676,469 | 9,840,000 | 295 | 93,901,888 | 517,102,425 |
| TheSyndicateProject | 83,505 | 9,770,000 | 130 | 1,131,839 | 30,788,584 |
| stampylonghead | 232,990 | 9,580,000 | 192 | 1,250,025 | 40,696,418 |
| LipaoGamer | 795,995 | 9,480,000 | 1171 | 29,035,335 | 349,073,188 |
| Muselk | 978,803 | 9,360,000 | 249 | 14,428,208 | 460,542,677 |
| Kuplinov ► Play | 1,475,181 | 9,150,000 | 553 | 26,141,479 | 414,847,752 |
| Поззи | 3,048,233 | 9,080,000 | 674 | 37,150,340 | 707,215,939 |
| JP Plays | 827,188 | 9,040,000 | 519 | 28,803,442 | 431,843,285 |
| LOUD | 3,743,364 | 9,010,000 | 370 | 148,715,306 | 1,041,223,364 |
| Denis—Roblox & More! | 1,321,871 | 8,770,000 | 320 | 6,511,320 | 291,863,488 |
| Hueva | 555,262 | 8,710,000 | 165 | 17,542,253 | 342,766,579 |
| aLexBY11 | 333,076 | 8,610,000 | 235 | 12,204,058 | 105,569,748 |
| Slogo | 1,766,567 | 8,590,000 | 734 | 88,388,909 | 1,286,072,065 |
| BRKsEDU | 335,698 | 8,560,000 | 426 | 10,249,397 | 101,967,294 |
| Аид [VyacheslavOO] | 2,832,833 | 8,530,000 | 445 | 39,845,318 | 592,223,812 |
| Yair17 | 1,862,427 | 8,350,000 | 171 | 47,146,001 | 345,798,627 |
| Dynamo Gaming | 269,460 | 8,330,000 | 317 | 30,721,269 | 341,615,666 |
| TheBrainDit | 132,285 | 8,330,000 | 567 | 2,694,914 | 44,092,622 |
| Games EduUu | 1,047,474 | 8,300,000 | 75 | 20,625,081 | 186,404,936 |
| elrubius | 1,717,383 | 8,260,000 | 157 | 51,355,921 | 608,099,694 |
| FaZe Clan | 335,169 | 8,230,000 | 84 | 5,232,440 | 160,424,585 |
| Unspeakable | 2,200,326 | 8,080,000 | 140 | 26,188,531 | 1,068,316,293 |
| UDiEX2 | 831,378 | 8,050,000 | 259 | 17,466,081 | 358,366,168 |
| FrontaL Gaming | 845,313 | 8,000,000 | 149 | 13,462,363 | 338,815,543 |
| Rodrigo F Gamer Respeita | 590,935 | 7,990,000 | 65 | 20,393,228 | 120,608,185 |
| Levinho | 1,738,040 | 7,900,000 | 426 | 41,955,083 | 785,640,540 |

| | | | | |
|---|---|---|---|---|
| Atro/أترو | 8,503,731 | 7,800,000 | 412 | 92,912,040 | 844,352,826 |
| sTaXxCraft | 108,447 | 7,790,000 | 93 | 4,096,013 | 42,479,035 |
| CRACKS | 1,034,627 | 7,710,000 | 136 | 20,053,140 | 299,312,298 |
| BoomSniper | 316,036 | 7,680,000 | 89 | 11,971,070 | 203,567,553 |
| BanderitaX | 1,958,206 | 7,640,000 | 201 | 42,709,594 | 438,281,775 |
| JoshDub | 671,786 | 7,600,000 | 70 | 16,357,649 | 527,984,685 |
| Fe4RLess | 400,067 | 7,590,000 | 2 | 2,027,461 | 56,042,889 |
| I AM WILDCAT | 277,968 | 7,510,000 | 324 | 9,704,065 | 310,834,044 |
| GAME OVER مصطفى | 1,228,132 | 7,510,000 | 89 | 17,666,936 | 183,044,904 |
| Smosh Games | 99,121 | 7,430,000 | 92 | 1,697,993 | 41,208,716 |
| CGGG | 346,029 | 7,330,000 | 252 | 11,125,872 | 231,189,496 |
| MrLololoshka (Роман Фильченков) | 1,658,327 | 7,240,000 | 496 | 27,303,590 | 290,652,894 |
| YOGSCAST Lewis & Simon | 189,011 | 7,200,000 | 312 | 3,855,665 | 89,761,221 |
| Coffi Channel | 1,170,609 | 7,190,000 | 732 | 25,398,197 | 337,203,373 |
| EdisonPts | 3,980,809 | 7,190,000 | 368 | 39,847,448 | 814,252,289 |
| Daithi De Nogla | 248,926 | 7,070,000 | 308 | 6,110,740 | 133,410,463 |
| MoonKase | 1,057,066 | 7,040,000 | 811 | 29,981,015 | 368,686,906 |
| BETO GAMER | 313,547 | 7,030,000 | 436 | 19,643,384 | 312,804,676 |
| Canal Clash War | 2,192,117 | 6,940,000 | 358 | 36,374,881 | 223,099,286 |
| kemas pake z | 783,246 | 6,930,000 | 286 | 6,724,187 | 153,045,168 |
| DeGoBooM | 792,026 | 6,840,000 | 371 | 13,263,460 | 273,500,157 |
| iHasCupquake | 163,633 | 6,770,000 | 275 | 1,380,019 | 35,101,269 |

[a] Number of comments (total): 'Sum of direct comments (first level) and sub-comments (second level) on posts' (Fanpage Karma Academy). [b] Fans: 'Number of subscribers to a cannel' (Fanpage Karma Academy). [c] Number of publications: 'Number of videos published in the selected time period' (Fanpage Karma Academy). [d] Number of Likes: 'Average number of 'likes' on videos published in the selected period, divided by the number of videos in the selected period' (Fanpage Karma Academy). [e] Sum of the impressions of individual messages: 'Number of views of videos published in the selected period' (Fanpage Karma Academy).

(**a**)

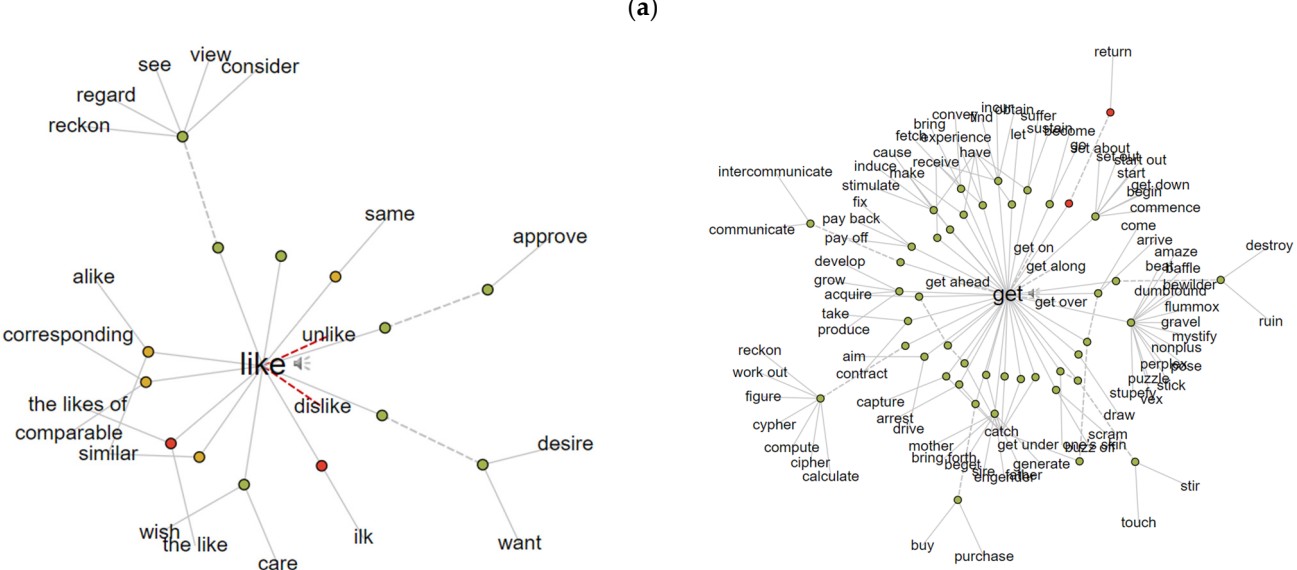

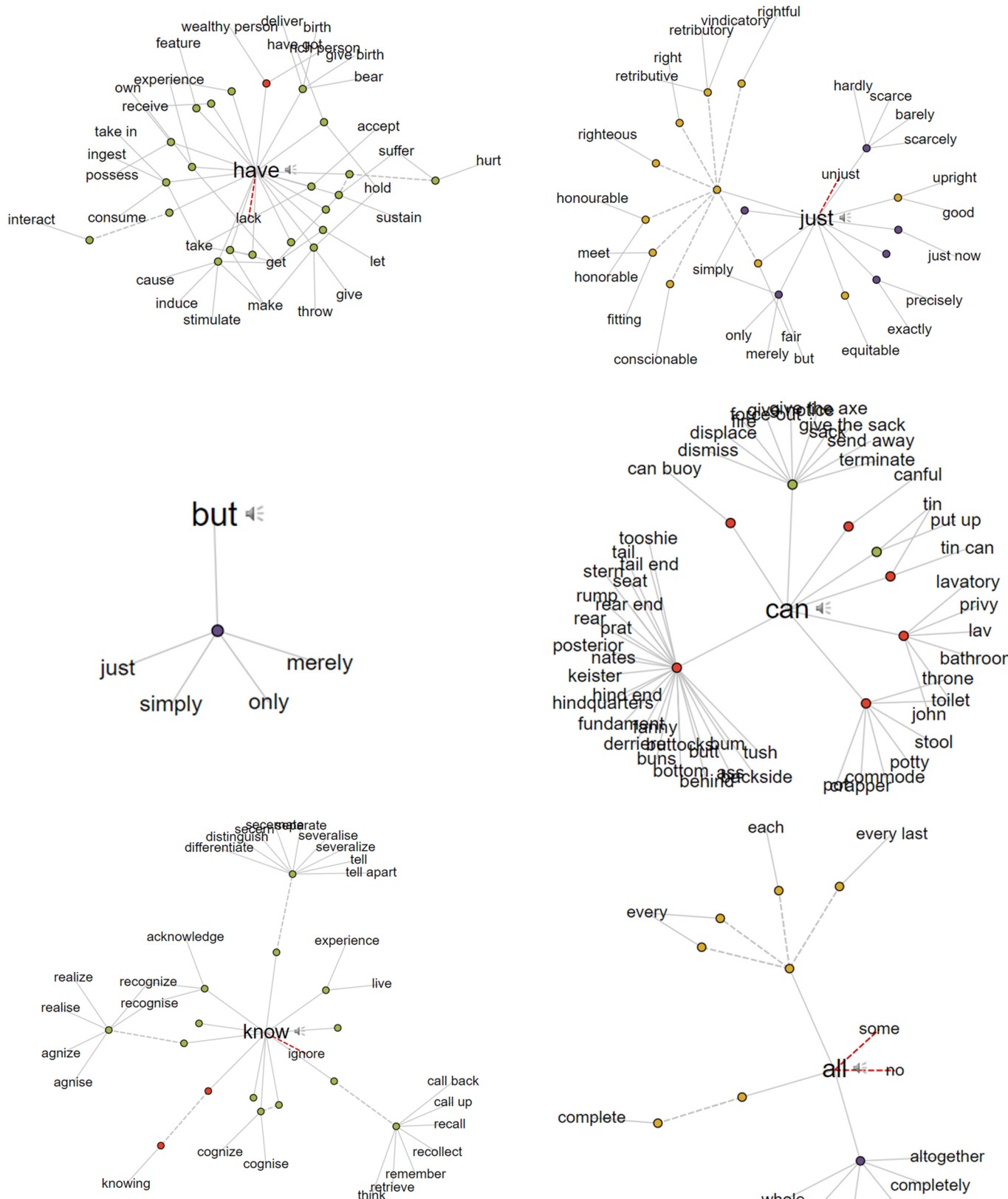

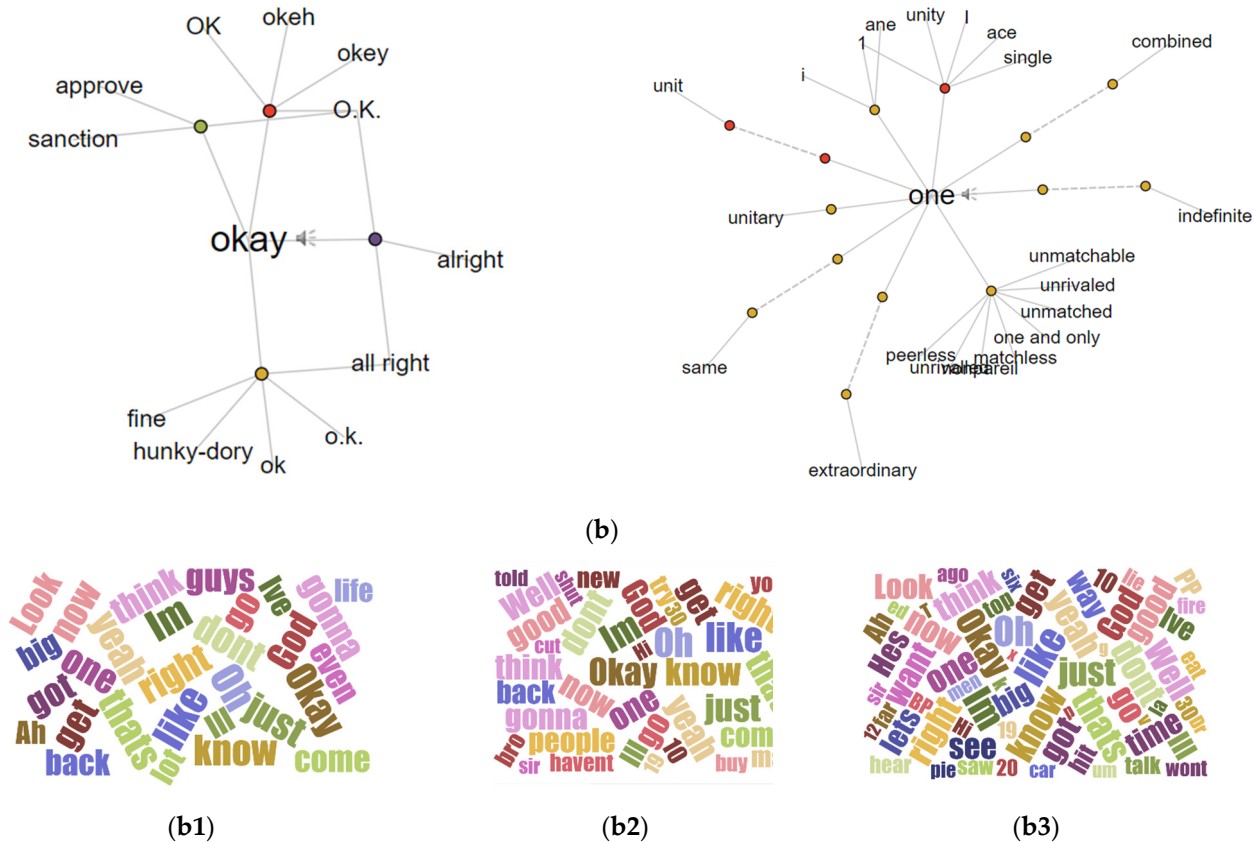

(**b**)

(**b1**)          (**b2**)          (**b3**)

**Figure A1.** Total semantic analysis of the 100 videos with the largest numbers of likes from 20 August 2019 to 20 August 2020 of the top 100 influencing gamers. (**a**) Branch graphics. (**b**) Density clouds. (**b1**) Density clouds with 100 most frequent words. (**b2**) Density clouds with 500 most frequent words. (**b3**) Density clouds with 1000 most frequent words.

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
