# Peer review of "Analizing Teens an Analysis from the Perspective of Gamers in Youtube"

_sustainability, doi:10.3390/su132011391_

Round 1
Reviewer 1 Report
The introduction and the materials and methods section are very dense. The information is very compressed. More detail is needed to facilitate understanding.
Nothing is dedicated to the purpose or relationship of the analysis to education. Some lines appear in the conclusions section but without showing the relationship or why.
It would be convenient to explain or clarify the application of the analysis, look for some reference on the didactic videos that appear on YouTube. Compare these videos with those of the gamers. Are the characteristics of the gamer videos the same as the didactic ones?
Page 10.- The numbering of the reference Pereira et al. it is not written correctly.
Page 11.- On this page there is a paragraph entirely in Spanish
Reviewer 2 Report
I attached a word file with the suggestions

Round 2
Reviewer 2 Report
Dear authors, I found the changes made very useful and helped me to better understand yours results. I believe that now the information contained in the paper and the readability are improved. Thanks for your work.
I have only one important point and some minor comments (typos I found in the text):
In the point 11 in your response, where I suggested to analyze the data from August 2019 to January 2020, you said that "Unfortunately, Fanpage Karma software is paid software, and our university only had money to study these actions for one year. If we have more funding again we would love to study what you propose."
I think that I did not make myself clear: what I meant is that, using the data that you already have, you can perform a robustness check (with meaning cloud, visual thesaurus and regression models) cutting the data at January 2020 (before pandemic, only 6 months instead that one year). This analysis could be useful to understand if there is some sort of "pandemic-effect". This is something that can make your results more robust.
Minor comments:
- in Instrument used and their validation " In other words, a sentiment analysis is performed. In other words, it analyzes the discourse..." replace the second "in other words" (maybe with "Specifically,"?)
- In KPI analysis "See Anex 2 to see escriptive results of the KPIs of...", miss D in descriptive.
- in Table 1 "..emotion, level..." delete the comma and after "agreement" there is a semicolon instead that a comma
- Annex 1, column "Gender of influencer", change all the "male/female" with the capital letter "Male/Female"
Thank you again for your work. One suggestion: when send out a paper, please, check thousand of times the text, searching for minor typos.
Author Response
Dear Editor and Reviewer,
Thank you for your help. I have made all the changes you have requested. I am sorry to send it back to you so quickly. I have to catch a plane for a European project meeting and then several trains and buses through mountainous area. I didn't want to run out of coverage.
Best regards
